# Diversity Among Clinical and Fresh Produce Isolates of *Stenotrophomonas*: Insights Through a One Health Perspective

**DOI:** 10.3390/foods15010023

**Published:** 2025-12-22

**Authors:** Alberto Pintor-Cora, Ángel Alegría, Ramiro López-Medrano, Jose M. Rodríguez-Calleja, Jesús A. Santos

**Affiliations:** 1Department of Food Hygiene and Food Technology, Veterinary Faculty, Universidad de León, 24071 León, Spain; aaleg@unileon.es (Á.A.); jm.rcalleja@unileon.es (J.M.R.-C.); j.santos@unileon.es (J.A.S.); 2Servicio de Microbiología Clínica, Complejo Asistencial Universitario de León, Gerencia Regional de Salud de Castilla y León (SACYL), 24008 León, Spain

**Keywords:** *Stenotrophomonas*, fresh produce, irrigation water, biofilms, antibiotic resistance, One Health

## Abstract

Fresh produce represents a key interface in the One Health continuum, connecting environmental, agricultural and clinical settings where opportunistic bacteria can circulate. Among them, *Stenotrophomonas* comprises an environmental genus of growing concern due to its multidrug resistance and rising clinical relevance. To investigate their diversity and pathogenic potential, nineteen isolates from vegetables, irrigation water and hospital sources were characterized by MLST, growth kinetics, biofilm formation, antimicrobial susceptibility assays and whole-genome sequencing. Phylogenetic analyses grouped 12 isolates within the *Stenotrophomonas maltophilia* complex (SMC) (clinical *S. maltophilia* (n = 7) and environmental *S. geniculata* (n = 4) and *S. sepilia* (n = 1)) and seven non-SMC isolates, including *S. indicatrix* (n = 5) and two unclassified clinical strains. Environmental *S. geniculata* and *S. sepilia* isolates showed robust growth at 37 °C and biofilm formation comparable to clinical lineages. Genomic analyses further revealed shared mobile loci (*afaD*, *fhaB*, *zot*) and homologous plasmids between environmental and clinical isolates, suggesting a connected gene pool. The identification of environmental strains with virulence-associated traits and clinical-like phenotypes supports fresh produce as a potential reservoir and transmission route for opportunistic *Stenotrophomonas*, underscoring the need for integrated surveillance across the food–health interface.

## 1. Introduction

Fresh produce serves as a critical interface within a One Health framework, facilitating the convergence of microorganisms from environmental, anthropogenic and animal reservoirs. Given its frequent raw consumption, the microbiota present on vegetable surfaces can be directly transferred to humans, where it may colonize both the skin and gastrointestinal tract. This enables dissemination within the community and healthcare settings through both direct and indirect pathways.

The genus *Stenotrophomonas* is associated with a wide range of habitats, including vegetables. In the environment, it holds ecological significance due to its involvement in cycling key elements. Nevertheless, certain species within the genus are recognized as opportunistic pathogens, commonly implicated in respiratory infections, soft tissue infections, urinary tract infections and bacteremia [1]. These infections predominantly affect individuals with underlying conditions such as cystic fibrosis, severe burns, cancer and various states of immunosuppression [2,3]. Moreover, it is frequently found as a coinfecting agent in respiratory viral illnesses such as COVID-19 [4,5]. Reported case-fatality rates are notably high, ranging from 20% to 69% [6,7].

The role of *Stenotrophomonas* in human disease has raised attention in recent years, leading to its recognition as an emerging pathogen. This trend is attributed to an increasing prevalence of vulnerability factors in the population, as well as the displacement of other pathogens such as *Pseudomonas* spp. and *Staphylococcus aureus*. This phenomenon is enhanced by the multidrug resistance profile of *Stenotrophomonas* and the selective pressure exerted by prolonged antibiotic use [8,9].

*Stenotrophomonas* occasionally colonizes animals or humans by exposure to environmental sources or contaminated medical equipment [10]. However, the pathways through which this pathogen enters the clinical setting or is directly transmitted to patients have not been extensively studied. Species of *Stenotrophomonas* have been isolated from different food commodities and from food contact surfaces, thus foodborne dissemination of pathogenic strains must be considered. Although *Stenotrophomonas* species are part of the plant-associated microbiota and are prevalent in agricultural environments, studies specifically investigating fresh produce as a vehicle for this opportunistic pathogen remain scarce [11].

The taxonomy of the *Stenotrophomonas* genus has undergone substantial revision in recent years. Within the genus, the *Stenotrophomonas maltophilia* complex (SMC) currently comprises the species *S. beteli*, *S. geniculata*, *S. hibiscicola*, *S. maltophilia*, *S. muris*, *S. pavanii*, *S. riyadhensis*, *S. sepilia* and the newly described *S. forensis*, being *S. maltophilia*, *S. geniculata* and *S. sepilia* the most dominant species [12]. These species are closely related, which complicates their accurate identification using common clinical diagnostic technologies such as MALDI-TOF MS, 16S rRNA gene sequencing or conventional phenotypic methods. As a result, a large proportion of clinical isolates are misidentified as *S. maltophilia*, underestimating the role of other environmentally prevalent species (e.g., *S. geniculata*, *S. sepilia*) in human infections [13,14].

Due to its opportunistic nature, the pathogenicity of *Stenotrophomonas* depends more on host vulnerability than on the presence of factors for tissue invasion, damage or immune evasion. Indeed, this genus is often regarded as a colonizer rather than an aggressive pathogen, particularly in the respiratory tract [15,16]. *Stenotrophomonas* strains share a conserved core repertoire of nonspecific virulence traits (including adhesins, flagella, siderophores, degradative enzymes and multiple secretion systems) which facilitate adherence and nutrient acquisition [17,18]. In addition, its intrinsic capacity to form biofilms plays a central role in its pathogenic potential by enabling long-term persistence and establishment in tissues and indwelling medical devices [19].

Another notable attribute of the *Stenotrophomonas* genus is its broad intrinsic antibiotic resistance profile. This resistance is mainly linked to the production of two β-lactamases (L1 and L2), which confer resistance to all β-lactam antibiotics except ceftazidime, as well as to aminoglycoside-modifying enzymes, multiple efflux pumps and reduced membrane permeability. This scenario greatly complicates the treatment of infections and limits therapeutic options to certain fluoroquinolones and the combination of sulfamethoxazole-trimethoprim. In addition to its intrinsic resistance profile, the acquisition of specific gene elements related to resistance to antimicrobial agents, including mobile elements carrying *qnr* and *sul* genes, has been described, which threatens the future management of this opportunistic pathogen [20,21].

This conserved toolbox, together with the limited characterization of well-defined virulence determinants, makes it difficult to distinguish between pathogenic and non-pathogenic strains, as almost any isolate may cause disease under favorable conditions [14,17,22,23].

This work was carried out to study the presence of *Stenotrophomonas* in fresh produce and to evaluate the genetic diversity, pathogenic potential and antimicrobial resistance of isolates obtained from fresh produce and clinical samples using an integrated phenotypic and genomic approach.

## 2. Materials and Methods

### 2.1. Sample Collection and Detection of Stenotrophomonas Isolates

Food-related isolates were obtained from fresh vegetable samples collected in farms and street markets located in the province of León, northwest Spain. Sampling campaigns took place during summer, autumn and early spring between July 2020 and September 2021 to ensure representation across different seasonal conditions. Vegetable samples included lettuce (*Lactuca sativa*), tomato (*Solanum lycopersicum*), cucumber (*Cucumis sativus*), carrot (*Daucus carota subsp. sativus*), frisee (*Cichorium endivia* var. *latifolium*), pepper (*Capsicum annuum*), parsley (*Petroselinum crispum*) and coriander (*Coriandrum sativum*). Additional samples from farm environments were also studied (soil, irrigation water, air and worker hands).

Ten grams of each vegetable and soil sample were homogenized with 90 mL of buffered peptone water (BPW; Oxoid, Thermofisher, Basingstoke, UK). Water samples were processed by filtering 100 mL of the sample through a 0.45 μm filter and the filter was then soaked in 100 mL of BPW. One hand swab sample was taken from each farm worker and the swab was then placed in a 10 mL tube containing BPW. Homogenates were incubated for 24 h at 37 °C. One loopful of the enriched solution was streaked onto ChromAgar ESBL (ChromAgar, Paris, France) and ChromAgar KPC (ChromAgar) for the detection of antibiotic resistant bacteria.

Nine clinical isolates were obtained from a first-level hospital (Complejo Asistencial Universitario de León-CAULE, León, Spain). Isolation sources included the respiratory tract (n = 5), soft tissues (n = 2) and abdominal cavity (n = 2).

Suspected isolates were identified as belonging to the genus *Stenotrophomonas* by matrix-assisted laser desorption/ionization-time of flight mass spectrometry (MALDI-TOF MS; Bruker Daltonics, Bremen, Germany).

### 2.2. Multilocus Sequence Typing (MLST) and Analysis

Multilocus sequence typing (MLST) was carried out using the scheme designed by Kaiser et al. [24] for isolates belonging to the *Stenotrophomonas* genus, which included seven genes: *atpD*, *gapA*, *guaA*, *mutM*, *nuoD*, *ppsA*, *recA*. Each gene was amplified independently by PCR, purified (GFX PCR DNA and Gel Band Purification Kit, Cytiva, Marlborough, MA, USA) and sequenced. The sequences were compared with the sequences from PUBMLST database [25] to obtain allele number and sequence type (ST).

Phylogenetic analysis was conducted by aligning a concatenation of the seven housekeeping genes and the phylogenetic trees were constructed using the neighbor-joining algorithm with the distances estimated by the maximum composite likelihood model and a bootstrapping of 1000 replications using MEGA11 software [26]. The reference genomes of *Stenotrophomonas* species available on RefSeq database were included for identification purposes.

### 2.3. Bacterial Growth Curve Determination at 37 °C

*Stenotrophomonas* isolates were cultured overnight in Tryptone Soya Broth (TSB; 30 °C, 140 rpm), adjusted to an OD_550_ of 1.000 ± 0.05 and diluted 1:100 in fresh broth prior to use. Two hundred µL of the diluted cultures were transferred into wells of polystyrene microtiter plates (Corning, Corning, NY, USA). Plates were incubated for 24 h at 37 °C and OD_600_ was recorded in a SpectraMax ID3 plate reader (Molecular Devices, San Jose, CA, USA). Each isolate was tested in quadruplicate and experiments were repeated in three independent assays.

### 2.4. Assessment of Biofilm Production Through Crystal Violet Assay

Two hundred µL of diluted cultures of *Stenotrophomonas* isolates prepared as described above were transferred into wells of polystyrene microtiter plates (Corning). Plates were incubated for 24 h at 37 °C and 22 °C in a sealed humidity-controlled chamber where 3 mL of sterile water were placed in the base of the container to minimize evaporation and prevent the well-edge effect. Negative control consisted of 200 μL of TSB.

Following incubation, cultures were removed from the wells and washed twice with 200 µL of phosphate-buffered saline (PBS) to remove non-adhered cells. Biofilms were fixed by incubating the plates at 60 °C for 1 h and stained with 200 µL of 0.1% crystal violet solution for 15 min. Crystal violet was removed and plates were washed twice with 200 µL of PBS. Plates were left to air dry for 30 min before the stain was solubilized with 200 µL of 33% acetic acid. OD_570_ of the solution was measured in a SpectraMax ID3 plate reader. Each isolate was tested in quadruplicate and experiments were repeated in three independent assays.

OD_570_ values were processed in accordance with Stepanović et al. [27]. Mean value of the isolates was calculated from the four replicates and ODc value was subtracted. ODc value was calculated from the mean value of OD_570_ of the negative control wells. Isolates were classified as strong (OD > ODc × 4), moderate (ODc × 4 > OD > ODc × 2), weak (ODc × 2 < OD) and no biofilm producers (ODc > OD).

### 2.5. Antimicrobial Susceptibility Testing

Antimicrobial susceptibility testing was carried out by disk-diffusion method following EUCAST guidelines. Antibiotic selection included the following agents: ceftazidime (CAZ, 30 µg), cefotaxime (CTX, 30 µg), cefepime (CPM, 30 µg), meropenem (MEM, 10 µg), gentamicin (GM, 10 µg), levofloxacin (LV, 5 µg), tigecycline (TGC, 15 µg), trimethoprim-sulfamethoxazole (SXT; 1.25/23.75 µg), minocycline (MN, 30 µg) and chloramphenicol (CHL, 30 µg).

Minimum inhibitory concentration (MIC) against colistin was established by agar dilution method using Colistin-ADATAB (MAST Group, Bootle, UK). Briefly, 1 µL of a 0.5 McFarland-adjusted solution of isolates was spotted on Mueller Hinton plates containing two-fold dilutions of colistin at concentrations ranging from 2 to 8 μg/mL. Plates were incubated for 24 h at 37 °C. MIC was established as the lowest concentration of the antimicrobial which visibly inhibited the growth of each isolate.

### 2.6. Genetic Characterization by Whole Genome Sequencing

Whole genome sequencing was conducted in a selected group of strains (CI04E2, CI11E, ZA29E2, AG41E, LE16E1, HSM1, HSM6, HSM7 and HSM8) based on the phenotypic and phylogenetic characteristics shown in previous assays. Genomic DNA was extracted using Wizard bacterial DNA extraction kit and quantified using Qubit fluorometer (Thermo Fisher Scientific, Waltham, MA, USA). Bacterial genomes were sequenced using Illumina technology. Read assembly was carried out using the BV-BRC tools (https://www.bv-brc.org/). Assembled contigs were annotated using the RAST online platform (https://rast.nmpdr.org).

Pairwise average nucleotide identity (ANIb) comparisons were performed among the isolates and reference genomes available in the NCBI database using JSpeciesWS [28].

Virulence factor detection was performed according to the set of target genes proposed by Adamek et al. [17] and Wang et al. [29]. Protein sequences for these genes were retrieved from UniProt(release 2025_049) and BLASTed against the genomes using the Seed Viewer tool in RAST, with manual curation of the results. The reference amino acid sequences for the virulence determinants are provided in Appendix A. Antimicrobial resistance genes were identified using the NCBI AMRFinderPlus tool (v4.0.23; database release 2025-06-03.1) with default settings. Genomic islands and acquired elements were identified using IslandCompare (https://islandcompare.ca/, accessed on 20 September 2025) and manually inspected using SnapGene Viewer (www.snapgene.com) and BLASTp (NCBI).

### 2.7. Data Processing and Statistical Analysis

Growth curves at 37 °C were analyzed using the Growthcurver package in RStudio (v. 2025.04). OD_600_ measurements over time were fitted to a logistic model to estimate the growth rate (r), carrying capacity (k) and area under the curve (auc). Only curves with a residual standard deviation (sigma) below 0.05 were included to ensure accurate fitting.

Biofilm production (OD_570_) was analyzed using non-parametric tests due to non-normal data distribution. Differences among isolates were assessed by Kruskal–Wallis test followed by Dunn’s post hoc test with Bonferroni correction to identify homogeneous groups. Pairwise comparisons of biofilm production at 22 °C and 37 °C for each isolate were performed using the Wilcoxon rank-sum test. Outliers were identified by boxplot inspection and removed based on the interquartile range (IQR) method. Statistical analyses and plots were performed in R (R Core Team).

Heatmaps of pairwise ANIb comparisons were generated using Heatmapper (www.heatmapper.ca, accessed on 10 September 2025) [30], whereas clustering figures and phylogenetic trees were constructed with ChipPlot (www.chiplot.online, accessed on 30 september 2025) [31].

## 3. Results

### 3.1. Isolation and Taxonomic Classification of Stenotrophomonas *spp.* Isolates

A total of 145 samples of vegetables and 90 environmental samples were analyzed. Sampling of vegetables and their production environment led to the isolation of a total of 46 colonies from ChromAgar KPC and 26 colonies from ChromAgar ESBL. Among these suspected β-lactamase-producing isolates, 10 strains were identified as belonging to the *Stenotrophomonas* genus through MALDI-TOF analysis. Eight were obtained from vegetable samples (coriander, lettuce, pepper, parsley and tomato), one from irrigation water and one from the hands of a farm worker. MALDI-TOF classified five of them as *S. maltophilia*, the remaining five isolates were classified as *Stenotrophomonas* sp. All clinical isolates were identified as *S. maltophilia* except for HSM5, which was classified as *S. rhizophila* (Table 1).

### 3.2. MLST Analysis

The phylogenetic analysis of the concatenated seven housekeeping genes resulted in the tree shown in Figure 1. Complete allelic profiles for each isolate can be found in Appendix A. The isolates clustered into two major groups: SMC strains and non-SMC strains.

Within the SMC strains, 7 clinical strains were clustered as *S. maltophilia* (HSM1, HSM2, HSM3, HSM4, HSM7, HSM8 and HSM9), while environmental strains were classified as *S. geniculata* (n = 4; LE42E1, CI11E, CI04E2 and ZA29E2) and *S. sepilia* (n = 1; AG41E). The non-SMC was composed of five environmental *S. indicatrix* strains (MA16E2, LE16E1, PJ15E, PI29E2 and TO26E) and two clinical strains which did not show significant proximity with any reference sequence (HSM5 and HSM6).

Most of the isolates related to vegetables and irrigation water had novel sequence types with low allelic correspondence to previously described STs. The only exception was the waterborne *S. sepilia* strain AG41E which showed high similarity with ST966, with only 5 nucleotide substitutions in *guaA*.

All clinical strains could be assigned to previously described sequence types: ST31 (n = 2; HSM1, HSM8), ST5 (n = 2; HSM3 and HSM7); ST115 (n = 1; HSM2); and ST172 (n = 2; HSM4 and HSM9); except for HSM5 and HSM6, which belonged to novel STs.

### 3.3. Average Nucleotide Identity (ANIb) Analysis

ANIb analysis was performed with the whole genome sequences of the isolates against reference genomes (Figure 2). The analysis indicated that 7 out of the 9 sequenced strains showed more than 95% identity with the reference genomes of previously described *Stenotrophomonas* species, while both HSM5 and HSM6 had nucleotide homologies below the species identification threshold (95%) against all reference strains. HSM5 showed the highest nucleotide identity with the genomes of *S. bentonitica*, *S. nematodicola* and *S. rhizophila* (83–85%), whereas HSM6 showed the closest similarity to *S. indicatrix* and *S. lactitubi* species (88%).

### 3.4. Growth Kinetics at 37 °C

The *Stenotrophomonas* strains were classified into four distinct groups based on growth parameters (*r*, *k* and *auc*) and the overall profile of their growth curves (Figure 3). HSM8 was identified as the top performer (Group A), exhibiting markedly elevated values of carrying capacity and area under the curve (*k* = 1.5; *auc* = 19.4). The remaining SMC strains, except HSM3 and HSM7, were assigned to Group B, displaying efficient and consistent growth kinetics (*k* = 0.8–1; *auc* = 12.2–14.7). In contrast, SMC strains HSM3 and HSM7 showed limited growth and were clustered with environmental *S. indicatrix* strains and the clinical isolate HSM6, forming Group C (*k* = 0.4–0.7; *auc* = 6.6–9.8). HSM5 exhibited no measurable growth and was therefore classified separately as Group D, characterized by a flat growth curve. Growth models and strain grouping can be found in Figure 3.

### 3.5. Biofilm-Forming Ability

*Stenotrophomonas* isolates were classified (Figure 4) as strong (OD > 0.6), moderate (0.3 < OD ≤ 0.6), weak (0.15 < OD ≤ 0.3), or non-biofilm producers (OD ≤ 0.15). SMC isolates showed strong biofilm production at 37 °C. At 22 °C most SMC isolates maintained or increased biofilm biomass (HSM1, HSM2, HSM8, CI11E, LE42E1, CI04E2, ZA29E2 and AG41E), whereas HSM4 and HSM9 lost detectable biofilm. Non-SMC isolates (HSM6, LE16E1, PJ15E, PI29E2, TO26E and HSM5) were weak or non-producers at both temperatures, except *S. indicatrix* MA16E2, which showed markedly increased biofilm formation at 22 °C and behaved as a strong producer.

### 3.6. Antimicrobial Susceptibility Profiles of *Stenotrophomonas* Isolates

The antimicrobial susceptibility profiles revealed considerable phenotypic variation among *Stenotrophomonas* isolates, which was consistent with their phylogenetic clustering (Figure 5).

Among β-lactam antibiotics, the isolates displayed a uniform resistance pattern to cefotaxime and meropenem, except for the waterborne *S. sepilia* isolate AG41E and the clinical isolates HSM6 and HSM5. In contrast, the resistance pattern to ceftazidime was more heterogeneous, with seven isolates (*S. maltophilia* HSM1 and HSM8, all *S. geniculata* (n = 4) and *S. indicatrix* LE16E1 isolates) showing a distinct resistance pattern to this agent.

Regarding aminoglycosides and polymyxins, most SMC isolates exhibited reduced susceptibility to gentamicin and colistin, as indicated by inhibition zones below 20 mm or MIC values above 8 μg/mL. HSM3 was the only exception, showing lower tolerance to both agents (28 mm for gentamicin; MIC < 2 μg/mL for colistin). Resistance to chloramphenicol was observed in *S. geniculata* strains and in *S. maltophilia* HSM8. HSM8 was also the only isolate resistant to sulfamethoxazole-trimethoprim, levofloxacin and tigecycline, in addition to chloramphenicol. By contrast, HSM5 remained fully susceptible to all antibiotics tested.

### 3.7. Detection of Virulence and Antibiotic Resistance Genes

Virulence factors in the genomes of the sequenced strains were examined according to the schemes proposed by Adamek [17] and Wang [29] and the results are shown in Figure 6.

The extracellular enzymes (proteases *smptr1* and *smptr2*, *katA*, siderophore *EntA*) and the hemolysin *hlyIII* were present in all isolates. However, the distribution of biofilm-related factors differed between SMC and non-SMC strains. Among the SMC, the biofilm-related genes *fliC*, *spgM*, *papD*, *pilU* and *rmlA* were detected in all strains, regardless of clinical or environmental origin. In contrast, in the non-SMC strains (LE16E1, HSM5 and HSM6), *smf-1* was not detected in any and *papD* and *fliC* were also absent in LE16E1.

The non-fimbrial adhesin *afaD* was detected in the clinical isolates HSM1 and HSM8 and in the irrigation water isolate AG41E. This gene was located next to a tRNA, flanked by hypothetical proteins and RHS proteins. The containing region showed a GC content of 62% in *S. maltophilia* and 57% in *S. sepilia*, compared to an average of 65% in the surrounding genome, supporting its mobile nature. The filamentous hemagglutinin *fhaB* was present in the same four isolates (HSM1, HSM8 and AG41E) and additionally in HSM5 and was also located within a genomic island.

The toxin *Zot* was detected exclusively in AG41E and was integrated within a complete prophage which shared 65.4% homology with the previously described *Stenotrophomonas* filamentous phage phiSMA6, which also carried the *zot* gene [32].

AMRFinderPlus analysis revealed that all SMC isolates harbored variants of the *blaL1* and *blaL2* genes, as well as aminoglycoside-modifying enzymes including *aph(6)*, *aph(3)* and *aph(9)*. In addition, *aac(6′)-Iz* was exclusively detected in strain HSM7. Among the non-SMC isolates, HSM6 and LE16E1 carried both *blaL1* and *blaL2*, as well as *aph(3)* and *aph(9)*, but lacked *aph(6)*. In contrast, in strain HSM5 only *blaL2* could be detected and none of the *aph* genes were present. Resistance genes associated with trimethoprim-sulfamethoxazole and fluoroquinolone resistance (*qnrB1*, *qnrB2*, *qnrB71*, *sul1*, *sul2* and *dfrA* genes) were not detected in any of the strains analyzed.

Finally, plasmids with high homology (>80%) were found in strains HSM8, CI11E and CI04E2. These plasmids encoded proteins associated with type II and type IV secretion systems, with no antibiotic resistance genes detected.

## 4. Discussion

The taxonomy of *Stenotrophomonas* is complex and has been revised in recent years, with an increasing number of new species described over the past decade [14,33,34,35]. Such taxonomic complexity has led to the definition of the *Stenotrophomonas maltophilia* complex (SMC), which groups together phylogenetically related species with significant clinical relevance as human pathogens [36]. Although the specific composition of the SMC is still undefined and several genomospecies are yet to be described, the most up-to-date interpretation of the complex includes at least *S. beteli*, *S. geniculata*, *S. forensis*, *S. hibiscicola*, *S. maltophilia*, *S. muris*, *S. pavanii*, *S. riyadhensis* and *S. sepilia* [12].

In this study, MALDI-TOF was used for the identification of the isolates, while MLST and ANIb were employed for species-level confirmation [37,38]. MALDI-TOF identification achieved 100% accuracy for clinical *S. maltophilia* isolates (n = 7); however, environmental isolates assigned to *S. geniculata* (n = 4) and *S. sepilia* (n = 1) by MLST and ANIb were identified as *S. maltophilia* by MALDI-TOF. Regarding species outside the *S. maltophilia* complex (SMC), *S. indicatrix* (n = 5) could only be identified at species level by MLST. Two clinical isolates, HSM5 (identified as *S. rhizophila* through MALDI-TOF) and HSM6 (identified as *Stenotrophomonas* sp. through MALDI-TOF), could not be assigned to any known species, as they showed Average Nucleotide Identity (ANI) values below the 95% threshold proposed for species delineation within the *Stenotrophomonas* complex [14].

Identification using standard clinical diagnostic methods, such as commercial MALDI-TOF databases or 16S gene sequencing, has proven insufficient for accurate species-level classification of *Stenotrophomonas* isolates [14,39]. As shown in our results, species within the SMC are often identified as *S. maltophilia* due to their close phylogenetic relationships and the absence of other species in the ID library and many can only be reliably distinguished by approaches such as MLST or whole-genome sequencing. A recent genomic analysis found that up to 58.6% of strains initially classified as *S. maltophilia* were misclassified [13].

As previously mentioned, 7 out of 9 clinical isolates were grouped within the SMC, all of which were identified as *S. maltophilia*. Among the ten environmental isolates five were classified within the SMC: one *S. sepilia* isolate from irrigation water and four *S. geniculata* isolates obtained from various vegetables (lettuce, coriander and carrot). Both *S. geniculata* and *S. sepilia* are species commonly found in environmental settings, having been detected in wastewater, soil and plant-associated environments [40,41,42,43,44], but they are also frequently detected in cases of human infections, each contributing approximately 10% of infections [13,29]. Their presence in both clinical and environmental contexts suggests a connection between these settings and underscores their potential role as emerging opportunistic pathogens [3,5,13,45].

Outside of the SMC, five environmental isolates were classified as *Stenotrophomonas indicatrix*. Furthermore, clinical strain HSM6, isolated from a soft tissue infection, clustered within the *S. indicatrix/lactitubi* group and showed the highest ANIb value (88%) compared to reference genomes of both species. *S. indicatrix* is an environmental species detected in soil, rhizosphere [46,47] and food contact industrial surfaces [48]. To the best of our knowledge, *S. indicatrix* is infrequently isolated from human clinical samples and is generally regarded as a pathogen of limited clinical relevance. The reported cases of human infection are restricted to a strain obtained from the respiratory tract and a genomic analysis which reclassified five clinical strains as *S. indicatrix* [13,49].

Clinical strain HSM5 could not be classified at the species level. This isolate was obtained from soft tissue infection and clustered within the *rhizophila*/*bentonitica/nematodicola* group through MLST-based identification showing ANIb values ranging from 83–85% against the reference genomes of these species. Species within this cluster are not commonly associated with clinical environments or human pathogenicity and are typically found in soil and rhizosphere environments, even being proposed as biotechnological agents. Until recently, these species have been considered non-pathogenic [23,50]. Authors such as Li et al. and Ochoa-Sánchez et al. report, just as in this study, human clinical strains classified within sister clades of *S. rhizophila* that originally were identified as *S. maltophilia* [13,36]. This information suggests the possible existence of novel entities related to this cluster that are likely associated with plant-related environments and could have clinical relevance.

Our findings highlight the need to refine species-level classification within the genus *Stenotrophomonas*. Accurate taxonomic resolution is essential to reveal the true epidemiology of both SMC and non-SMC species and understand the routes involved in its transmission to humans [13]. Optimizing identification technologies and reaching consensus on species delineation are critical to avoid misclassification, which otherwise complicates cross-study comparisons and hinders the detection of emerging pathogenic species other than *S. maltophilia* [35,36].

All of the environmental isolates were typified as novel sequence types, except for the irrigation water isolate *S. sepilia* AG41E, which showed an almost exact match with sequence type ST966. ST966 has been previously detected in lake waters in China by Zhang et al. (2024) [51]. This lineage demonstrated high virulence in in vitro cell culture assays and *Galleria mellonella* infection models, along with a strong biofilm-forming capacity [51]. The detection of virulent clones in water underscores the role of irrigation as a critical pathway for the introduction of pathogenic and antibiotic-resistant bacteria into the agricultural environment.

Clinical *S. maltophilia* strains could be assigned to previously described sequence types: ST31 (n = 2; HSM1 and HSM8), ST115 (n = 1; HSM2), ST5 (n = 2; HSM3 and HSM7) and ST172 (n = 2; HSM4 and HSM9). All of them have been previously isolated from human infections (www.pubmlst.org). Among them, ST5 is frequently reported as the most common sequence type in hospital infections in different studies, being detected in several countries including USA, France, Austria, Korea, China, Spain and Italy [52,53,54,55,56]. This lineage appears to be widely disseminated in clinical settings and well adapted to the human host. ST31 has also been detected in several studies [53,55,56]. Clinical strains HSM5 and HSM6 belonged to novel sequence types, thus their clinical relevance cannot be discussed.

The complexity of virulence within the *Stenotrophomonas* genus, its likely multifactorial nature and the lack of clearly defined virulence factors in the literature make it essential to assess its pathogenicity from a phenotypic perspective [2]. In this work, we analyzed three features proposed as key determinants of the clinical significance of *Stenotrophomonas* isolates: growth at 37 °C, biofilm formation and their antimicrobial susceptibility profile.

The ability to grow at 37 °C is an essential trait for the human pathogenic potential of a bacterium. Among the isolates analyzed in this study, members of the SMC complex showed good growth capacity at this temperature, with comparable growth rates between environmental and clinical isolates. The only exception was the group formed by the clinical isolates HSM3 and HSM7, which belonged to the same sequence type (ST5) and exhibited lower growth rates compared to the other SMC isolates and comparable to that of *S. indicatrix* isolates in this study. In contrast, the isolate HSM8 stood out by displaying a much higher growth rate than the rest of the SMC. The inability to grow at this temperature is a feature commonly associated with strictly phytosymbiont species such as *S. rhizophila*, supporting their presumed non-pathogenic nature for humans [57]. However, in this study we report a non-SMC isolate, HSM5, which was involved in a human soft tissue infection despite being unable to grow at 37 °C. This observation indicates that the inability to grow at 37 °C is not always sufficient to rule out the pathogenic potential of *Stenotrophomonas*. The involvement of HSM5 in an infection suggests that certain non-SMC lineages may still exploit specific ecological niches in the host, such as wounds or superficial tissues where local temperatures are lower than core body temperature and therefore may retain clinical relevance even without robust growth at 37 °C.

The capacity to produce biofilm on polystyrene was evaluated at 37 °C and at 22 °C. All isolates belonging to the SMC showed a strong capacity to form biofilms on polystyrene at 37 °C. Biofilm formation is considered an attribute widely spread across the SMC [18,19,58]. The reduction in incubation temperature to 22 °C to emulate environmental conditions resulted in variable responses among SMC isolates. Most of the SMC strains maintained (CI04E2, LE42E1 and HSM8) or even increased (HSM1, HSM2, HSM9, CI11E and AG41E) their biofilm accumulation at 22 °C, in some cases up to three-fold. However, two isolates, HSM4 and HSM9 (both belonging to ST172), showed a marked reduction, shifting from strong biofilm producers to no detectable biofilm formation. This suggests the presence of specific regulatory mechanisms or genetic traits that may rely on gene expression or pathway activation at 37 °C. The strong ability to form biofilms at both 37 °C and 22 °C detected in the strains isolated from fresh produce differs from the findings of Klimkaite et al. (2025), who reported that approximately 70% of environmental isolates were weak or non-producers at both 37 °C and 28 °C, whereas about 90% of clinical isolates formed biofilms under the same conditions [59].

In contrast, the non-SMC strains, including the environmental *S. indicatrix* isolates and the clinical isolates HSM5 and HSM6, generally failed to form biofilms at either temperature, being classified as weak or non-producers. The only exception was *S. indicatrix* MA16E2, which exhibited moderate ability at 37 °C and markedly increased biofilm formation at 22 °C, shifting to a strong producer at this temperature.

The ability to form biofilms represents a major advantage for *Stenotrophomonas* persistence and dissemination across the food chain. On vegetables, biofilm formation promotes survival on the product surface and limits removal by standard washing or disinfection methods. Biofilms also favor bacterial accumulation during post-harvest processing, thereby facilitating persistence in food production environments and increasing the likelihood of transmission to humans [60]. The strong biofilm-forming capacity observed in some vegetable-associated SMC isolates indicates not only their ability to persist along the food chain, but also their potential to establish themselves as opportunistic pathogens once introduced into nosocomial environments [58].

Biofilm formation in the genus *Stenotrophomonas* remains poorly understood, involving the interplay of numerous genes as well as complex regulatory networks, such as those mediated by the DSF system [18,19,61]. In this study, all SMC isolates harbored a conserved set of genes associated with biofilm formation, including genes related to initial adhesion (*smf-1*, *pilU*, *fliC*) and exopolysaccharide production (*rmlA*, *spgM*). This observation is consistent with previous reports which indicate that these factors are widespread across the complex [18,29,59,62]. In fact, genomic analyses suggest that most of the described virulence factors do not differ significantly between environmental and clinical strains [17,37,59]. This supports the hypothesis that the strains from the SMC harbor a stable core set of virulence-associated genes, particularly those linked to biofilm formation, that may enable opportunistic pathogenic potential throughout the SMC.

Beyond this conserved toolbox, some SMC isolates also harbored additional virulence factors located within acquired genomic islands. The non-fimbrial adhesin *afaD* was detected in clinical isolates *S. maltophilia* HSM1 and HSM8, as well as in the irrigation water isolate *S. sepilia* AG41E. This gene encodes a surface-associated protein that mediates attachment of *Stenotrophomonas* to host cells, initial adhesion and subsequent biofilm formation [29,62]. Wang et al. (2024) report a high but heterogeneous prevalence of this factor in clinical isolates of *S. maltophilia* and *S. sepilia*, with detection rates of 84% and 71%, respectively [29]. However, it appears to be much less frequent in other species within the SMC, such as *S. geniculata* [29].

The filamentous hemagglutinin *fhaB* was also detected in a mobile element in HSM1, HSM8, AG41E and interestingly, in HSM5. Filamentous hemagglutinins are large proteins encoded in repeat sequences that play an important role in systemic infection due to their function as hemagglutinins, promoting cell aggregation and hemolysis; but also because of their implications in the adherence to and invasion of host cells [63]. The *fhaB* gene is a factor detected sporadically in clinical isolates of *S. maltophilia*, with no extensive data available on its prevalence. When it is detected, it is described to be included in genomic islands, supporting its acquired and mobile nature [17,62,63].

All sequenced non-SMC isolates (LE16E1, HSM5, HSM6) lacked the *smf-1* gene and additional adhesion factors (*papD*, *fliC*) were also absent in *S. indicatrix* LE16E1. As *smf-1* plays an important role in initial adhesion and biofilm formation, its absence could contribute to the weak or absent biofilm phenotypes observed in these isolates [64,65]. Nevertheless, clinical isolates HSM5 and HSM6 may present alternative adhesion mechanisms, such as the filamentous hemagglutinin detected in HSM5 [62,63]. At the same time, previous studies suggest that absence of *smf-1* reduces biofilm formation but can enhance acute virulence traits, as both functions appear to be inversely regulated [64,66,67]. Although biofilm formation is a major determinant of environmental persistence and chronic infections, alternative mechanisms may sustain pathogenicity in acute infections, which remain poorly characterized in *Stenotrophomonas* and are largely inferred from chronic respiratory disease models [68].

Several additional virulence factors were consistently identified across the isolates. Serine proteases such as *smptr1* and *smptr2* have been described as key factors in the cytotoxic effects of the genus, exhibiting a broad substrate specificity that enables them to degrade the extracellular matrix and damage host cell structures [69]. In our study, both factors were detected in 100% of SMC strains, including both clinical and environmental. In contrast, the available literature reports lower and more heterogeneous prevalence rates for the protease *smptr1* [29,53,62], particularly in environmental isolates [53,59]. The hemolysin *hlyIII* was likewise detected in all sequenced isolates, supporting its role as a conserved virulence determinant within the genus.

In contrast, the putative toxin Zot was detected exclusively in the waterborne isolate *S. sepilia* AG41E. The *zot* locus was located within a complete prophage showing 65.4% identity to the filamentous phage phiSMA6, also carrying *zot* and capable of integrating into *S. maltophilia* genomes [32], supporting a phage-mediated acquisition and dissemination of this factor. Prevalence of *zot* in *S. maltophilia* clinical isolates is heterogeneous, ranging from undetected [62] to 24% [59]. These findings may indicate an environmental origin of the virulence factor and highlight the evolutionary role of bacteriophages as drivers of gene transfer within the genus *Stenotrophomonas* [23].

Although its function in *Stenotrophomonas* is still unclear, it has been proposed as the first true virulence factor described in this genus [32]. In *Vibrio cholerae*, Zot acts as an enterotoxin by disrupting intestinal tight junctions and increasing epithelial permeability [70]. This protein shared conserved features with homologs in human pathogens as *V. cholerae* (25%) or *Campylobacter concisus* (25%), including a 216-residue N-terminal domain [32,71]. Further functional studies are needed to clarify its role in pathogenesis and infection severity.

Further evidence of horizontally acquired elements was provided by a 60 kb plasmid detected in *S. maltophilia* HSM8, which encoded proteins of both the type II (T2SS) and type IV secretion systems (T4SS) but no known antibiotic resistance determinants. This plasmid constituted the major genomic difference with the closely related isolate HSM1 (>99% ANI) and showed >80% homology to plasmids from environmental *S. geniculata* CI11E and CI04E2, also carrying secretion systems, suggesting the existence of a shared gene pool between environmental and clinical *Stenotrophomonas* populations.

The T4SS is a multifunctional apparatus involved in both effector delivery and horizontal gene transfer [72]. In human infections, it displays dual activity: delaying apoptosis to maintain host cell viability while inducing apoptosis in immune cells such as macrophages, thereby promoting immune evasion [73]. At the same time, the T4SS provides competitive advantages by killing cohabiting species like *Pseudomonas*, a frequent co-colonizer in respiratory polymicrobial infections [69,73,74]. The retention of multiple divergent T4SS clusters in our isolates likely reflects functional specialization, enhancing bacterial fitness and potentially contributing to disease severity [72].

The evaluation of antibiotic resistance in *Stenotrophomonas* remains a significant challenge [75,76]. Factors such as biofilm formation can confer increased tolerance to antimicrobial drugs, making it difficult to accurately predict the true resistance of isolates of this genus to these agents during treatment [18]. Moreover, there are currently no established breakpoints for many of the drugs used to treat *Stenotrophomonas* infections, which complicates both clinical decision-making and the interpretation of susceptibility testing for surveillance purposes.

The *Stenotrophomonas* isolates exhibited a homogeneous resistance pattern to cefotaxime and carbapenems, which can be attributed to the production of the intrinsic β-lactamases L1 (metallo-β-lactamase) and L2 (class A serine-β-lactamase) characteristic of the genus. The exceptions to this trend were the waterborne *S. sepilia* isolate AG41E and clinical HSM5 and HSM6 which appeared to be less tolerant to meropenem, cefotaxime and ceftazidime.

Among β-lactam antibiotics, ceftazidime has been commonly employed for the treatment of *Stenotrophomonas* infections due to the susceptibility to this agent despite the production of intrinsic β-lactamases, as it is a weak inducer of L2 production [77,78]. In this study, 30% of clinical *S. maltophilia* isolates showed resistance to ceftazidime, with only the ST31 isolates (HSM1 and HSM8) showing resistance to the antibiotic. Of particular note is the pattern observed among the fresh produce isolates of *S. geniculata*, which seem to be intrinsically resistant to this agent [13]. Resistance rates to ceftazidime have progressively increased over time and recent reports indicate that 40–60% of clinical *S. maltophilia* isolates remain susceptible [29,79]. Ceftazidime is currently not recommended as a first-line treatment option and more reliable alternatives such as sulfamethoxazole-trimethoprim (SXT), levofloxacin (LEV), tigecycline (TGC) or minocycline (MN) are preferred [33].

In addition to ceftazidime, *S. geniculata* isolates consistently showed resistance to chloramphenicol, further supporting the existence of intrinsic species-specific resistance profiles [13]. These findings underscore the importance of accurate taxonomic identification for guiding appropriate therapeutic approaches.

Tolerance to gentamicin was only observed among SMC isolates and was absent in non-SMC strains. This phenotype can be attributed to genomic differences, as SMC isolates consistently carried a broader set of aminoglycoside-modifying enzymes, including *aph(6)*, *aph(3)* and *aph(9)*. By contrast, non-SMC isolates displayed a more heterogeneous pattern: HSM5 lacked all three genes, whereas LE16E1 and HSM6 retained only *aph(3)* and *aph(9)* but not *aph(6)*. Similarly, colistin resistance appeared to be a consistent feature among all SMC isolates, reinforcing prior findings that discourage the use of colistin for *Stenotrophomonas* infections despite its effectiveness against other Gram-negative pathogens [80].

Only the clinical isolate *S. maltophilia* HSM8 showed resistance to critical antibiotics including sulfamethoxazole-trimethoprim, levofloxacin and tigecycline, with an inhibition halo diameter for SXT just above the EUCAST defined breakpoint (16 mm). Currently, prevalence data on isolates resistant to these agents in clinical settings vary considerably depending on the study and the geographic region analyzed, with reported ranges of 4–25% for SXT, 4–43% for LEV and 10–20% for TGC [8,16,52,77,81]. Although we did not observe resistance to these agents among the fresh produce isolates, previous studies have reported values within these ranges in environmental isolates of SMC [59].

Resistance to these agents can be conferred by the acquisition of specific resistance determinants such as multiple *qnr* variants, *sul1*, *sul2* and *dfrA* [82,83], none of which were detected in the genome of HSM8. Thus, this resistance pattern may reflect intrinsic broad-resistance mechanisms, mainly driven by efflux pump overexpression [36,84]. This hypothesis is further supported by the observation that, compared with its closely related strain HSM1 (ST31), HSM8 shows a phenotypic shift toward resistance to multiple agents, which is consistent with selection under antibiotic pressure in the nosocomial environment [85,86]. In *Stenotrophomonas*, a large efflux arsenal (up to eight RND, two ABC and two MFS systems) is a key driver of adaptive resistance to multiple toxicants, including those frequently present in agricultural environments such as pesticides, heavy metals and diverse phytochemicals. Continuous exposure to these agents may act as a training phase that selects for efflux overexpression, ultimately promoting cross-resistance in clinical settings [86].

The only antibiotic that showed effectiveness against all isolates was minocycline. This agent has been recommended, together with cefiderocol [1,79], as an alternative given the growing resistance to agents such as sulfamethoxazole-trimethoprim and levofloxacin and the concerning trends suggesting that resistance to these drugs will continue to increase.

This study provides evidence suggesting the potential pathogenicity and transmission of SMC isolates through fresh produce and plant-associated environments. In this work, we report the isolation of strains with genotypic virulence profiles similar to those found in human clinical isolates and in some cases, carrying additional acquired virulence factors [13]. This genetic evidence of virulence is further supported by their phenotypic profiles, as they were classified as strong biofilm producers with growth curves equivalent to clinical strains at 37 °C. The simultaneous presence of virulence factors and a strong capacity for biofilm formation underscores the need to pay closer attention to *S. sepilia* and *S. geniculata* strains originated in agricultural environments [13]. These strains could act not only as direct pathogens but also as potential vectors for the transmission of mobile virulence factors [87]. These findings demonstrate that fresh produce can act as a route for the introduction of *Stenotrophomonas* strains into vulnerable patients or nosocomial settings. Notably, this study detected strains with clinical implications (HSM5 and HSM6) not belonging to SMC. Although non-SMC strains have been sporadically reported in human infections, their actual contribution to disease emergence is likely underestimated [13,36].

## 5. Conclusions

This study, despite the limited sample size and restricted geographic coverage, underscores the need for broader surveillance to better characterize the diversity, pathogenic potential and antimicrobial resistance of this genus in agricultural production environments.

The results obtained suggest that the continuous interplay between environmental and anthropogenic settings may facilitate the exchange of *Stenotrophomonas* strains. Fresh produce appears to have the potential to act as a route for introducing environmental strains with pathogenic traits into clinical and community environments, whereas the use of contaminated aquifers and especially recycled urban water might enable the introduction of human-associated strains into agricultural systems. Such bidirectional movement could promote adaptation and favor the horizontal transfer of virulence and antimicrobial resistance genes between both settings.

Monitoring water sources is therefore essential not only to prevent the introduction and circulation of human-adapted strains in the food production chain but also as a critical control point to disrupt the cycle of adaptation and genetic exchange across environmental and clinical contexts. Strengthening measures to limit the transmission of *Stenotrophomonas* through fresh produce could help reduce the disease burden associated with this pathogen and mitigate the evolutionary pressures driving the acquisition of additional virulence and antimicrobial resistance factors.

## Figures and Tables

**Figure 1 foods-15-00023-f001:**
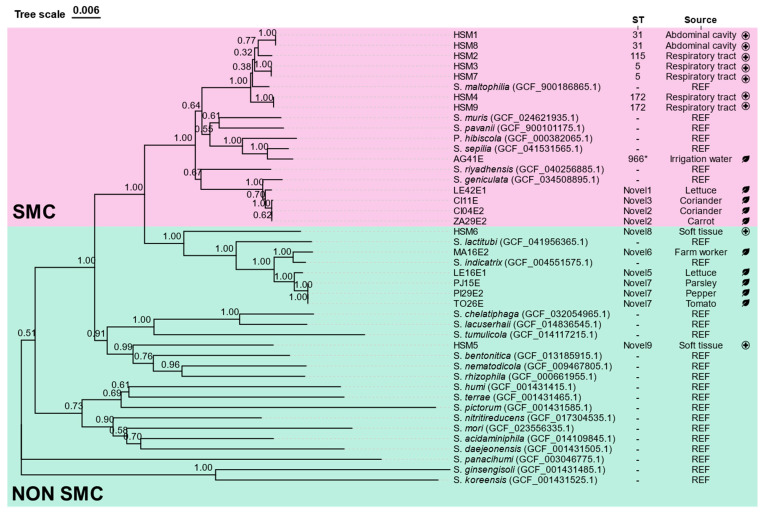
Phylogenetic tree based on the concatenated sequences of seven housekeeping genes. The ST and source of study isolates are indicated. The tree was inferred using the neighbor-joining algorithm. Reference (“REF”) genomes available for *Stenotrophomonas* species were included for identification purposes. The two major clades are shown in color: the SMC (*Stenotrophomonas maltophilia* complex) group (pink) and the non-SMC group (blue). Bootstrapping values are indicated at the nodes. * indicates 6/7 allelic coincidences with the ST included. Isolates from agricultural environments are represented by a leaf symbol, whereas isolates from clinical environments are represented by a cross.

**Figure 2 foods-15-00023-f002:**
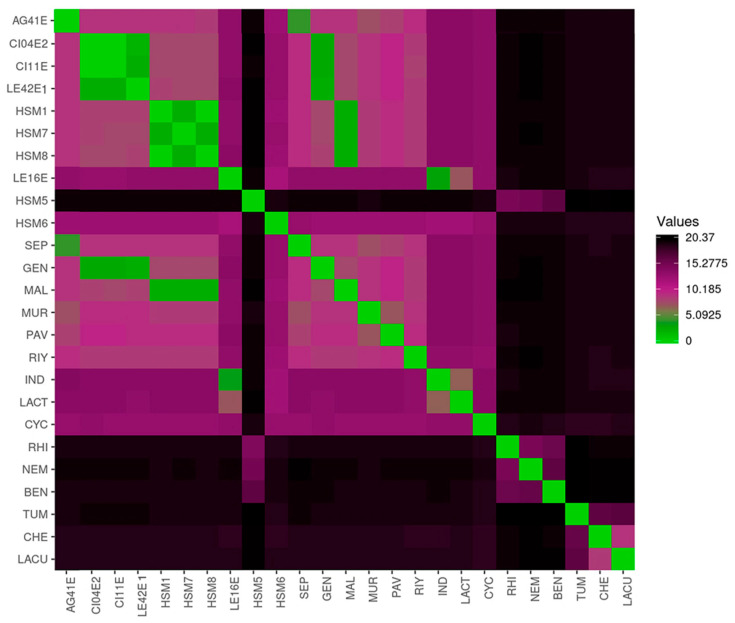
Heatmap showing pairwise distance values based on Average Nucleotide Identity (ANIb) among the *Stenotrophomonas* isolates and reference genomes. Each cell represents the calculated genomic distance between two genomes, with colors ranging from green (high similarity) to purple-black (lower similarity). Isolate codes are shown on both axes. The color scale bar indicates ANIb distance values from 0 (identical) to 20.37 (maximum observed distance). SEP: *S. sepilia*, GEN: *S. geniculata*, MAL: *S. maltophilia*, MUR: *S. muris*, PAV: *S. pavanii*, RIY: *S. riyadhensis*, IND: *S. indicatrix*, LACT: *S. lactitubi*, CYC: *S. cyclobalanopsidis*, RHI: *S. rhizophila*, NEM: *S. nematodicola*, BEN: *S. bentonitica*, TUM: *S. tumulicola*, CHE: *S. chelatiphaga*, LACU: *S. lacuserhaii*.

**Figure 3 foods-15-00023-f003:**
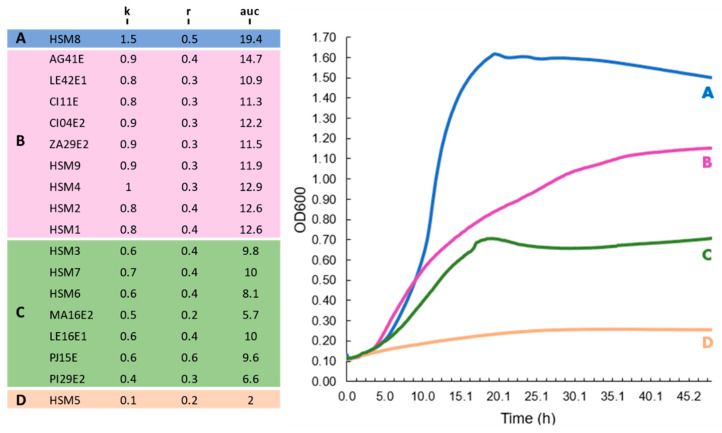
Growth dynamics of *Stenotrophomonas* strains at 37 °C. Isolates were clustered in 4 groups (**A**–**D**) based on carrying capacity (*k*), maximum growth rate (*r*) and area under the growth curve (*auc*). Representative growth curves (**right**) illustrate the average growth curve profiles of each group.

**Figure 4 foods-15-00023-f004:**
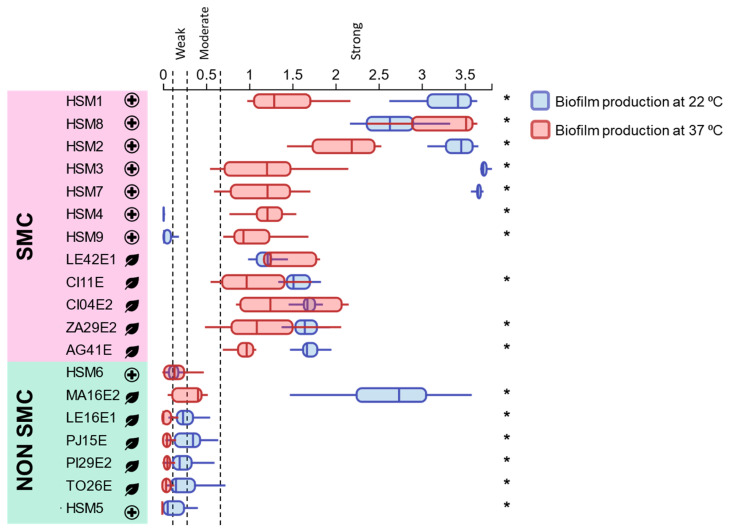
Biofilm-forming ability of the *Stenotrophomonas* isolates at 37 °C and 22 °C, measured by the crystal violet staining assay (OD_570_ values). Boxplots represent the OD_570_ values for the different replicates across the three independent experiments; the horizontal line within each box indicates the median. * indicates significant differences within the same isolate for biofilm production at 22 °C and 37 °C (*p* < 0.05). SMC: *Stenotrophomonas maltophilia* complex. Isolates from agricultural environments are represented by a leaf symbol, whereas isolates from clinical environments are represented by a cross.

**Figure 5 foods-15-00023-f005:**
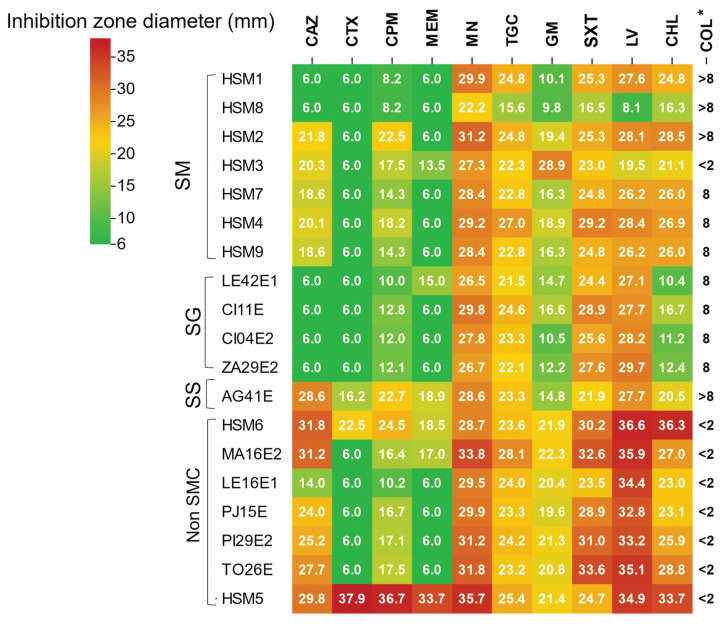
Antimicrobial susceptibility profile of *Stenotrophomonas*. Color scale is associated with the diameter of the inhibition zone (mm). CAZ: ceftazidime, CTX: cefotaxime, CPM: cefepime, MEM: meropenem, MN: minocycline, TGC: tigecycline, GM: gentamicin, SXT: trimethoprim-sulfamethoxazole, LV: levofloxacin, CHL: chloramphenicol, COL: colistin. * MIC value (μg/mL) is provided for colistin. SM: *Stenotrophomonas maltophilia*, SG: *Stenotrophomonas geniculata*, SS: *Stenotrophomonas sepilia*.

**Figure 6 foods-15-00023-f006:**
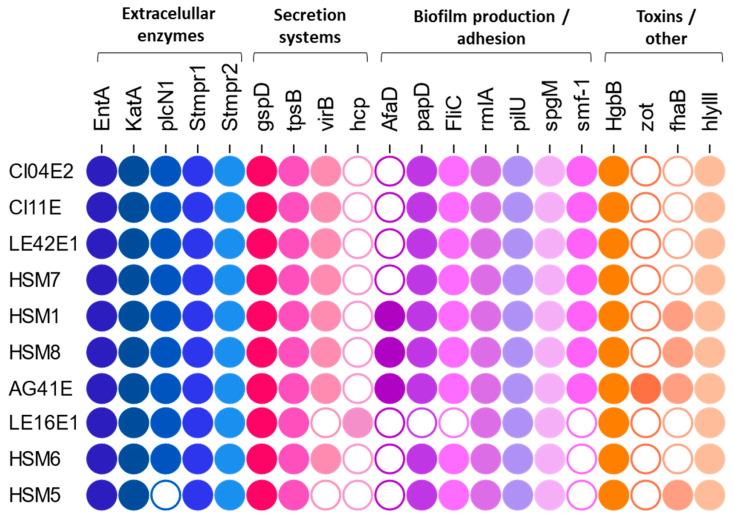
Distribution of virulence factors across the sequenced isolates. A filled circle (●) indicates presence, while an empty circle (○) indicates that the factor has not been detected in that isolate.

**Table 1 foods-15-00023-t001:** Taxonomic assignment of clinical and fresh produce *Stenotrophomonas* isolates by MALDI-TOF and MLSA/ANIb.

**Isolate**	**MALDI-TOF**	**MLSA/AniB**	**Source**	
AG41E	*Stenotrophomonas maltophilia*	*Stenotrophomonas sepilia*	Irrigation Water	Environmental/Fresh produce
CI04E2	*Stenotrophomonas maltophilia*	*Stenotrophomonas geniculata*	Coriander	
CI11E	*Stenotrophomonas maltophilia*	*Stenotrophomonas geniculata*	Coriander	
LE16E1	*Stenotrophomonas* sp.	*Stenotrophomonas indicatrix*	Lettuce	
LE42E1	*Stenotrophomonas maltophilia*	*Stenotrophomonas geniculata*	Lettuce	
MA16E2	*Stenotrophomonas* sp.	*Stenotrophomonas indicatrix*	Farm Worker	
PI29E2	*Stenotrophomonas* sp.	*Stenotrophomonas indicatrix*	Pepper	
PJ15E	*Stenotrophomonas* sp.	*Stenotrophomonas indicatrix*	Parsley	
TO26E	*Stenotrophomonas* sp.	*Stenotrophomonas indicatrix*	Tomato	
ZA29E2	*Stenotrophomonas maltophilia*	*Stenotrophomonas geniculata*	Carrot	
HSM1	*Stenotrophomonas maltophilia*	*Stenotrophomonas maltophilia*	Abdominal cavity	Clinical
HSM2	*Stenotrophomonas maltophilia*	*Stenotrophomonas maltophilia*	Respiratory tract	
HSM3	*Stenotrophomonas maltophilia*	*Stenotrophomonas maltophilia*	Respiratory tract	
HSM4	*Stenotrophomonas maltophilia*	*Stenotrophomonas maltophilia*	Respiratory tract	
HSM5	*Stenotrophomonas rhizophila*	*Stenotrophomonas* sp.	Soft tissue	
HSM6	*Stenotrophomonas maltophilia*	*Stenotrophomonas* sp.	Soft tissue	
HSM7	*Stenotrophomonas maltophilia*	*Stenotrophomonas maltophilia*	Respiratory tract	
HSM8	*Stenotrophomonas maltophilia*	*Stenotrophomonas maltophilia*	Abdominal cavity	
HSM9	*Stenotrophomonas maltophilia*	*Stenotrophomonas maltophilia*	Sputum	

## Data Availability

All genomic data produced in this work are publicly available in the European Nucleotide Archive (ENA) under the BioProject accession PRJEB104872, which includes raw sequencing reads, sample metadata and genome assemblies.

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
