# Peer review of "Diversity Among Clinical and Fresh Produce Isolates of *Stenotrophomonas*: Insights Through a One Health Perspective"

_foods, 2025, doi:10.3390/foods15010023_

Round 1

Reviewer 1 Report

Comments and Suggestions for Authors

This is an outstanding manuscript entitled "Diversity Among Clinical and Fresh Produce Isolates of Stenotrophomonas: Insights Through a One Health Perspective." The study presents a systematic comparison of Stenotrophomonas isolates from clinical and fresh produce sources using phenotypic and genomic approaches. It offers valuable insights into the transmission and pathogenic potential of this opportunistic pathogen within the "One Health" framework. The research is well-designed, meticulously executed, and clearly presented. It significantly advances our understanding of Stenotrophomonas evolution and transmission in a One Health context. I recommend acceptance after minor revisions to further solidify the evidence and conclusions.

  1. Line 17-19: The study analyzes a total of 19 strains, with a subset undergoing whole-genome sequencing. This relatively limited sample size may constrain the statistical power of the analysis and the generalizability of the conclusions. It is recommended that the authors acknowledge this limitation in the discussion section.
  2. Line528-537: The study compellingly infers the pathogenic potential of environmental isolates through genomic and phenotypic assays such as biofilm formation. To further strengthen the evidence, it would be valuable to include more direct virulence assessments in future work. For instance, adhesion/invasion assays using epithelial cell lines or employing infection models like Galleria mellonella could provide a direct comparison of virulence between clinical and environmental isolates.
  3. Line 377 and Line 593: Some findings that are already clearly presented in the Results section are reiterated in detail in the Discussion. Condensing this descriptive text would allow for a greater focus on interpreting the results, discussing their implications, and highlighting the broader significance of the study.
  4. Line 123: The use of ChromAgar ESBL/KPC for enrichment and isolation is a selective method that specifically targets bacteria carrying ESBL/KPC genes. This approach may have inadvertently introduced a selection bias by excluding Stenotrophomonas strains that lack these specific resistance determinants. It is important to acknowledge this potential bias in the discussion, as it may affect the representativeness of the collected strain panel and the generalizability of the findings regarding the overall population.
  5. Line 159: Regarding the biofilm assay, the authors mention taking measures to "prevent the well-edge effect caused by evaporation." To ensure the reproducibility of the experiment, could they please provide specific details on how humidity was controlled?
  6. Line 710: The clinical isolate HSM5 exhibited very limited growth capacity at 37°C. Given that it still caused an infection, this presents a fascinating paradox that warrants further discussion. The authors are encouraged to speculate on its potential pathogenic mechanism. For instance, could its success be highly dependent on a severely immunocompromised host, or might it rely on unique virulence factors not detected in this study?
  7. Line 314: An interesting observation is the significant difference in biofilm formation at 22°C versus 37°C for some clinical strains (e.g., HSM4 and HSM9). We suggest that the discussion would be greatly strengthened by speculating on the potential molecular mechanisms underlying this temperature-dependent regulation. For instance, could this be linked to differential expression of biofilm-related genes or regulatory systems (e.g., quorum-sensing) in response to temperature shifts?
  8. Fig 5: Strain HSM8 demonstrates resistance to multiple critical drugs (e.g., SXT, LV, TGC), yet no corresponding acquired resistance genes were detected. While the authors appropriately hypothesize that efflux pump overexpression could be a potential mechanism, providing experimental evidence would significantly strengthen this claim. Were any experiments conducted, or are there plans to use efflux pump inhibitors (such as CCCP or PABN) in combination with these antibiotics to assess susceptibility changes? Alternatively, comparative transcriptomics could be employed to validate the overexpression of specific efflux pump genes.
  9. Line 611: The discussion regarding the T2SS/T4SS identified on plasmids is insightful. However, it could be further deepened by expanding beyond their role in gene transfer. Specifically, the authors are encouraged to discuss how these secretion systems might directly contribute to virulence. For instance, could the T2SS be involved in secreting toxins or degradative enzymes, or could the T4SS function as a direct effector delivery system into host cells, akin to systems in other pathogens?

Author Response

We thank the reviewer for the evaluation of our manuscript and for the constructive comments provided. All changes suggested by this reviewer and the others have been incorporated into the revised manuscript, where they are highlighted in yellow. Below, we provide a point-by-point response to each comment.

Line 17-19: The study analyzes a total of 19 strains, with a subset undergoing whole-genome sequencing. This relatively limited sample size may constrain the statistical power of the analysis and the generalizability of the conclusions. It is recommended that the authors acknowledge this limitation in the discussion section.

Limitations have been acknowledged in the conclusion (Lines 721-724)

Line 528-537: The study compellingly infers the pathogenic potential of environmental isolates through genomic and phenotypic assays such as biofilm formation. To further strengthen the evidence, it would be valuable to include more direct virulence assessments in future work. For instance, adhesion/invasion assays using epithelial cell lines or employing infection models like Galleria mellonella could provide a direct comparison of virulence between clinical and environmental isolates.

We agree that incorporating direct virulence assays would substantially strengthen future analyses, and we already plan to include both epithelial cell adhesion/invasion assays and Galleria mellonella infection models in the next phase of this research.

Line 377 and Line 593: Some findings that are already clearly presented in the Results section are reiterated in detail in the Discussion. Condensing this descriptive text would allow for a greater focus on interpreting the results, discussing their implications, and highlighting the broader significance of the study.

The authors consider it important to highlight the presence of the toxin within the prophage-associated mobile element. Although this aspect is already mentioned in the Results section, its reiteration in the Discussion is intended to emphasize both the relevance of the finding itself and its potential implications for the pathogenicity of Stenotrophomonas. We agree that further condensation is possible and appreciate the suggestion, yet we believe that drawing attention to this genomic context provides essential background for interpreting the broader biological significance of the observation.

Line 123: The use of ChromAgar ESBL/KPC for enrichment and isolation is a selective method that specifically targets bacteria carrying ESBL/KPC genes. This approach may have inadvertently introduced a selection bias by excluding Stenotrophomonas strains that lack these specific resistance determinants. It is important to acknowledge this potential bias in the discussion, as it may affect the representativeness of the collected strain panel and the generalizability of the findings regarding the overall population.

The detection of this Stenotrophomonas collection was carried out within a broader project investigating the transmission of AMR bacteria through fresh produce, in which ChromAgar KPC was used as the primary medium, which is designed to detect multiple carbapenem-resistant bacteria. Stenotrophomonas species display intrinsic resistance to carbapenems due to the production of L1. Although ChromAgar KPC is not specifically designed for the isolation of Stenotrophomonas, the genus is not expected to experience growth inhibition and produces white colonies that can be selectively recognized. Notably, the recommended medium for Stenotrophomonas, VIA agar, also includes carbapenems in its formulation, which supports the presumption that carbapenem-containing media should not hinder its recovery.

Line 159: Regarding the biofilm assay, the authors mention taking measures to "prevent the well-edge effect caused by evaporation." To ensure the reproducibility of the experiment, could they please provide specific details on how humidity was controlled?

Methodology regarding the biofilm assay has been clarified in lines 158–160: “Plates were incubated for 24 hours at 37 ⁰C and 22 ⁰C in a sealed humidity-controlled chamber where 3 ml of sterile water were placed in the base of the container to minimize evaporation and prevent the well-edge effect”.

Line 710: The clinical isolate HSM5 exhibited very limited growth capacity at 37°C. Given that it still caused an infection, this presents a fascinating paradox that warrants further discussion. The authors are encouraged to speculate on its potential pathogenic mechanism. For instance, could its success be highly dependent on a severely immunocompromised host, or might it rely on unique virulence factors not detected in this study?

The discussion regarding HSM5, its inability to grow at 37 °C and its pathogenic potential has been expanded in lines 501–508.

Line 314: An interesting observation is the significant difference in biofilm formation at 22°C versus 37°C for some clinical strains (e.g., HSM4 and HSM9). We suggest that the discussion would be greatly strengthened by speculating on the potential molecular mechanisms underlying this temperature-dependent regulation. For instance, could this be linked to differential expression of biofilm-related genes or regulatory systems (e.g., quorum-sensing) in response to temperature shifts?

The temperature-dependent biofilm behavior observed in isolates such as HSM4 and HSM9 is indeed striking and suggests that underlying regulatory differences may exist. However, at this stage we are unable to explore potential molecular mechanisms in detail because complete genome sequences for these isolates are not yet available. We plan to address this question in future work through whole-genome sequencing and a broader comparative analysis.

Fig 5: Strain HSM8 demonstrates resistance to multiple critical drugs (e.g., SXT, LV, TGC), yet no corresponding acquired resistance genes were detected. While the authors appropriately hypothesize that efflux pump overexpression could be a potential mechanism, providing experimental evidence would significantly strengthen this claim. Were any experiments conducted, or are there plans to use efflux pump inhibitors (such as CCCP or PABN) in combination with these antibiotics to assess susceptibility changes? Alternatively, comparative transcriptomics could be employed to validate the overexpression of specific efflux pump genes.

No efflux pump inhibition assays or transcriptomic analyses were conducted as part of the present study, and these experiments are not planned within the current scope. Nonetheless, the reviewer’s comment is highly valuable and we fully agree that such approaches, would substantially strengthen the mechanistic understanding of resistance in HSM8 and other isolates that might present this resistance pattern. We intend to explore these possibilities in future work.

Line 611: The discussion regarding the T2SS/T4SS identified on plasmids is insightful. However, it could be further deepened by expanding beyond their role in gene transfer. Specifically, the authors are encouraged to discuss how these secretion systems might directly contribute to virulence. For instance, could the T2SS be involved in secreting toxins or degradative enzymes, or could the T4SS function as a direct effector delivery system into host cells, akin to systems in other pathogens?

The functional implications of the T2SS and T4SS are indeed of great relevance, yet the specific contribution of these systems to virulence in Stenotrophomonas remains insufficiently characterized. We discuss their putative roles based on functions described in other bacterial genera, since experimental evidence for Stenotrophomonas is still limited and the precise effectors delivered by these systems in this genus have not been fully elucidated.

Reviewer 2 Report

Comments and Suggestions for Authors

This research report presents a comparative study on the genetic diversity, virulence potential and antibiotic resistance of Stenotrophomonas strains isolated from fresh produce and clinical environments. It highlights the potential role of fresh produce as a reservoir and transmission route for opportunistic pathogens, which has significant public health implications. However, there are some areas for improvement in the depth of bioinformatics analysis and the interpretation of genomic findings in the paper.

  1. It is suggested that more detailed information such as the geographical location and season of sample collection be added in the Methods section to enhance the reproducibility and representativeness of the study.
  2. The manuscript does not mention uploading the raw genomic sequencing data to a public database (such as NCBI SRA) and providing the accession number. This is a common requirement and academic norm for publishing genomic-related research.
  3. All SMC strains have biofilm genes, but HSM4/HSM9 do not form biofilms at 22°C. The possible reasons should be analyzed in depth in the Discussion section.
  4. The clinical strains HSM5 and HSM6 cannot be classified into known species (ANI < 95%), and are only speculated to be new species. Further taxonomic analysis is needed.
  5. The Discussion section mentions that the SMC strains from fresh produce in this study have strong biofilm formation ability, which is different from the study by Klimkaite et al. in 2025 (70% of environmental strains are weak/non-toxigenic strains), but the reasons for the difference are not analyzed.
Comments on the Quality of English Language

This research report presents a comparative study on the genetic diversity, virulence potential and antibiotic resistance of Stenotrophomonas strains isolated from fresh produce and clinical environments. It highlights the potential role of fresh produce as a reservoir and transmission route for opportunistic pathogens, which has significant public health implications. However, there are some areas for improvement in the depth of bioinformatics analysis and the interpretation of genomic findings in the paper.

  1. It is suggested that more detailed information such as the geographical location and season of sample collection be added in the Methods section to enhance the reproducibility and representativeness of the study.
  2. The manuscript does not mention uploading the raw genomic sequencing data to a public database (such as NCBI SRA) and providing the accession number. This is a common requirement and academic norm for publishing genomic-related research.
  3. All SMC strains have biofilm genes, but HSM4/HSM9 do not form biofilms at 22°C. The possible reasons should be analyzed in depth in the Discussion section.
  4. The clinical strains HSM5 and HSM6 cannot be classified into known species (ANI < 95%), and are only speculated to be new species. Further taxonomic analysis is needed.
  5. The Discussion section mentions that the SMC strains from fresh produce in this study have strong biofilm formation ability, which is different from the study by Klimkaite et al. in 2025 (70% of environmental strains are weak/non-toxigenic strains), but the reasons for the difference are not analyzed.

Author Response

We thank the reviewer for the evaluation of our manuscript and for the comments provided. All changes suggested by this reviewer and the others have been incorporated into the revised manuscript, where they are highlighted in yellow. Below, we provide a point-by-point response to each comment.

  1. It is suggested that more detailed information such as the geographical location and season of sample collection be added in the Methods section to enhance the reproducibility and representativeness of the study.

Additional information regarding the geographical origin and sampling period has now been incorporated into the Methods section (line 110-113).

  1. The manuscript does not mention uploading the raw genomic sequencing data to a public database (such as NCBI SRA) and providing the accession number. This is a common requirement and academic norm for publishing genomic-related research.

Accession number included in the data availability statement

  1. All SMC strains have biofilm genes, but HSM4/HSM9 do not form biofilms at 22°C. The possible reasons should be analyzed in depth in the Discussion section.

The temperature-dependent biofilm behavior observed in isolates such as HSM4 and HSM9 suggests that underlying regulatory differences may exist. However, at this stage we are unable to explore potential molecular mechanisms in detail because complete genome sequences for these isolates are not yet available. We plan to address this question in future work through whole-genome sequencing and a broader comparative analysis.

  1. The clinical strains HSM5 and HSM6 cannot be classified into known species (ANI < 95%), and are only speculated to be new species. Further taxonomic analysis is needed.

We agree that the taxonomic interpretation of clinical isolates HSM5 and HSM6 requires caution. These strains were subjected to species-level discrimination using both MLSA and whole-genome ANI analyses, which represent the current gold-standard approach for taxonomic assignment in Stenotrophomonas. As shown in the manuscript, both isolates consistently fell below the 95% ANI threshold against all available reference genomes, indicating that they do not belong to any currently described species.

It should also be noted that the genus Stenotrophomonas contains a remarkably large proportion of genomically unresolved lineages: recent phylogenomic studies estimate that up to 40–60% of publicly available genomes cannot be assigned to any validly named species and instead cluster into multiple cryptic or yet-undescribed clades. In this context, the position of HSM5 and HSM6 is not unexpected and reflects the broader taxonomic complexity of the genus rather than an analytical limitation of this study.

  1. The Discussion section mentions that the SMC strains from fresh produce in this study have strong biofilm formation ability, which is different from the study by Klimkaite et al. in 2025 (70% of environmental strains are weak/non-toxigenic strains), but the reasons for the difference are not analyzed.

The discrepancy between our findings and those reported by Klimkaite et al. (2025) is difficult to interpret with certainty. The study by Klimkaite et al. did not classify isolates using species-level discriminatory methods, which may have led to grouping together members of the S. maltophilia complex (SMC) with species outside the complex, such as S. indicatrix, which are commonly associated with soil and, based on our results, tend to show limited biofilm-forming ability. Such combined analyses can obscure the true representativeness of SMC isolates, which are the lineages more frequently implicated in human infections, when evaluating their biofilm-forming capacity in environmental isolates.

Reviewer 3 Report

Comments and Suggestions for Authors

The manuscript presents a One-Health comparison of Stenotrophomonas isolates from vegetables, irrigation water, and clinical settings, integrating MLST, WGS, growth kinetics, biofilm assays, virulence gene profiling, and antimicrobial susceptibility testing. The research question is relevant and fits well within the scope of Foods, particularly as it concerns fresh produce as a vehicle for opportunistic pathogens.

The manuscript has clear strengths—particularly the multilayered phenotypic–genomic approach, but requires some revisions.

Major Comments

  1. The study uses ESBL and KPC selective media to isolate Stenotrophomonas, which is atypical and may bias the recovered population toward β-lactam-tolerant members.
    Please provide justification of selecting this choice.

Minor Comments

  1. Reduce redundancy when discussing MALDI misidentification.
  2. The manuscript discusses intrinsic L1/L2 β-lactamases, but a table linking phenotypes to the detected AMR genes could be also included.
  3. The manuscript could benefit from editing (some long or repetitive paragraphs).

Author Response

We thank the reviewer for the evaluation of our manuscript and for the comments provided. All changes suggested by this reviewer and the others have been incorporated into the revised manuscript, where they are highlighted in yellow. Below, we provide a point-by-point response to each comment.

Major Comments

  1. The study uses ESBL and KPC selective media to isolate Stenotrophomonas, which is atypical and may bias the recovered population toward β-lactam-tolerant members.
    Please provide justification of selecting this choice.

The detection of this Stenotrophomonas collection was carried out within a broader project investigating the transmission of AMR bacteria through fresh produce, in which ChromAgar KPC was used as the primary medium. Stenotrophomonas is assumed to display intrinsic resistance to carbapenems due to the production of L1. Although ChromAgar KPC is not specifically designed for the isolation of Stenotrophomonas, the genus is not expected to experience growth inhibition and typically produces white colonies that can be selectively recognized. Notably, the recommended medium for Stenotrophomonas, VIA agar, also includes carbapenems in its formulation, which supports the assumption that carbapenem-containing media should not hinder its recovery.

Minor Comments

  1. Reduce redundancy when discussing MALDI misidentification. The manuscript could benefit from editing (some long or repetitive paragraphs).

Thank you for this observation. We appreciate the reviewer’s suggestion and have carefully revisited the sections referring to MALDI-TOF misidentification. In our assessment, each mention appears in a different context (taxonomic challenges, methodological limitations and implications for surveillance), which is why some degree of reiteration was initially maintained. However, we agree that smoother narrative flow is desirable, and we have adjusted the text to avoid unnecessary repetition and to improve clarity while preserving the conceptual points that we consider essential for interpreting the results.

2. The manuscript discusses intrinsic L1/L2 β-lactamases, but a table linking phenotypes to the detected AMR genes could be also included

Only intrinsic Stenotrophomonas L1/L2 β-lactamase and quinolone resistance genes were detected in the isolates; therefore, their inclusion in an additional figure would have limited value in the context of this study and would result in excessive redundancy. The specific differences of interest are already highlighted and discussed in the Results (lines 381 to 388) and Discussion (lines 640-645 and 665-673). Moreover, the phenotypic patterns observed are more complex than simple presence–absence profiles, so a gene–phenotype table would not accurately reflect the underlying mechanisms and would not substantially enhance the manuscript.

Reviewer 4 Report

Comments and Suggestions for Authors

The manuscript examines diversity, phenotypic properties, antimicrobial resistance, and genomic characteristics of Stenotrophomonas isolates recovered from fresh produce, environmental samples, and clinical material within a  One Health framework. The study is timely, methodologically appropriate, and well aligned with the journal’s scope. However, several important issues require clarification, particularly regarding the consistency of MLST results, the interpretation of findings, and sampling-related limitations. 

Major comments

There are substantial inconsistencies in ST assignments across the Results, Discussion, and Supplementary Material. This undermines the reliability of the phylogenetic interpretation.
Please verify, correct, and unify all ST assignments throughout the manuscript  and provide a clear table listing each isolate together with its corresponding ST.

Given the relatively small number of isolates and their origin from a single gegraphic region, some conclusions regarding fresh produce as a reservoir or transmission route are expressed too strongly.
These claims should be toned down and presented as preliminary observations or hypotheses requiring further study.

Only 10 out of 19 isolates were subjected to whole-genome sequencing, yet the rationale for this selection is insufficiently explained.
Why were only 10 isolates chosen, and by what criteria? Please clarify the basis for WGS subset selection.

The manuscript appears to be limited by the sampling strategy, including its temporal and geographic constraints.
The authors should explicitly address these limitations and explain why broader sampling criteria were not implemented.

Minor comments:

Minor linguistic and editorial corrections are necessary (e.g., correcting “subtracted,” unifying antimicrobial names and gene formatting).

Formatting: ensure consistent use of °C and units (μL, μg/mL).

Replace all placeholder labels such as “Figure X” (e.g., in Section 3.4) with the correct figure numbers.

The rationale for using the Neighbor-Joining (NJ) method for phylogenetic reconstruction should be clarified.
Was a model test performed (e.g., using MEGA) to determine which evolutionary model best fits the dataset?

Author Response

We thank the reviewer for the evaluation of our manuscript and for the comments provided. All changes suggested by this reviewer and the others have been incorporated into the revised manuscript, where they are highlighted in yellow. Below, we provide a point-by-point response to each comment.

Major comments

There are substantial inconsistencies in ST assignments across the Results, Discussion, and Supplementary Material. This undermines the reliability of the phylogenetic interpretation.
Please verify, correct, and unify all ST assignments throughout the manuscript and provide a clear table listing each isolate together with its corresponding ST.

The typographical error has been corrected and the description of the ST distribution among isolates has been clarified to specify that isolates HSM1 and HSM8 belong to ST31, whereas HSM3 and HSM7 correspond to ST5.

Given the relatively small number of isolates and their origin from a single geographic region, some conclusions regarding fresh produce as a reservoir or transmission route are expressed too strongly. These claims should be toned down and presented as preliminary observations or hypotheses requiring further study.

We have now emphasized the preliminary nature of the findings in the Conclusions section (lines 721–741) and clarified that the present results serve as an initial foundation for elucidating the role of fresh produce as a potential vehicle for Stenotrophomonas transmission into nosocomial settings.

Only 10 out of 19 isolates were subjected to whole-genome sequencing, yet the rationale for this selection is insufficiently explained. Why were only 10 isolates chosen, and by what criteria? Please clarify the basis for WGS subset selection.

Due to practical and budgetary limitations, sequencing all 19 isolates was not feasible. Therefore, a multilayered selection strategy was implemented to maximize the representativeness of the WGS subset. The ten isolates were chosen to: (i) capture the phylogenetic diversity observed through MLST, (ii) include both clinical and non-clinical sources and (iii) represent different phenotypic profiles, particularly regarding antimicrobial susceptibility and biofilm formation.

The manuscript appears to be limited by the sampling strategy, including its temporal and geographic constraints. The authors should explicitly address these limitations and explain why broader sampling criteria were not implemented.

The present study was conceived as preliminary study focused on fresh-produce production sites and street markets within the province of León, which served as a representative case study to investigate the diversity, pathogenic potential and antimicrobial resistance of Stenotrophomonas associated with vegetables and farm environments. We acknowledge that a broader sampling strategy covering additional regions and extended time periods would increase the generalizability of the findings, yet such an approach was beyond the scope and logistical constraints of the current project. These limitations have now been explicitly acknowledged and incorporated into the Conclusions section (Lines 721– 741).

Minor comments:

Minor linguistic and editorial corrections are necessary (e.g., correcting “subtracted,” unifying antimicrobial names and gene formatting). Formatting: ensure consistent use of °C and units (μL, μg/mL). Replace all placeholder labels such as “Figure X” (e.g., in Section 3.4) with the correct figure numbers.

Formatting and editing mistakes have been now corrected. Thank you.

The rationale for using the Neighbor-Joining (NJ) method for phylogenetic reconstruction should be clarified. Was a model test performed (e.g., using MEGA) to determine which evolutionary model best fits the dataset?

Neighbor-Joining was selected as a robust and widely used distance-based method for MLST phylogenies, where the aim is strain-level discrimination rather than deep evolutionary inference. Distances were computed using the Maximum Composite Likelihood model implemented in MEGA, which accounts for unequal base frequencies and variable substitution rates. Alternative evolutionary models available in MEGA, including Kimura two-parameter, were also tested and produced no changes in the overall tree topology, supporting the stability of the inferred relationships.

Round 2

Reviewer 4 Report

Comments and Suggestions for Authors

The authors have made the most important revisions to the manuscript. Therefore, I believe it is now more valuable.